# Unidimensional Continuous-Variable Quantum Key Distribution with Untrusted Detection under Realistic Conditions

**DOI:** 10.3390/e21111100

**Published:** 2019-11-11

**Authors:** Luyu Huang, Yichen Zhang, Ziyang Chen, Song Yu

**Affiliations:** 1State Key Laboratory of Information Photonics and Optical Communications, Beijing University of Posts and Telecommunications, Beijing 100876, China; hly@bupt.edu.cn (L.H.); yusong@bupt.edu.cn (S.Y.); 2State Key Laboratory of Advanced Optical Communication, Systems and Networks, Department of Electronics, and Center for Quantum Information Technology, Peking University, Beijing 100871, China; chenziyang@pku.edu.cn

**Keywords:** unidimensional modulated coherent states, continuous-variable quantum key distribution, untrusted detection

## Abstract

A unidimensional continuous-variable quantum key distribution protocol with untrusted detection is proposed, where the two legitimate partners send unidimensional modulated or Gaussian-modulated coherent states to an untrusted third party, i.e., Charlie, to realize the measurement. Compared with the Gaussian-modulated coherent-state protocols, the unidimensional modulated protocols take the advantage of easy modulation, low cost, and only a small number of random numbers required. Security analysis shows that the proposed protocol cannot just defend all detectors side channels, but also achieve great performance under certain conditions. Specifically, three cases are discussed in detail, including using unidimensional modulated coherent states in Alice’s side, in Bob’s side, and in both sides under realistic conditions, respectively. Under the three conditions, we derive the expressions of the secret key rate and give the optimal gain parameters. It is found that the optimal performance of the protocol is achieved by using unidimensional modulated coherent states in both Alice’s and Bob’s side. The resulting protocol shows the potential for long-distance secure communication using the unidimensional quantum key distribution protocol with simple modulation method and untrusted detection under realistic conditions.

## 1. Introduction

Quantum key distribution (QKD) [1,2,3,4], as one of the most prominent applications of quantum information science, allows two legitimate partners, i.e., Alice and Bob, to achieve the secure key distribution phase of an encrypted communication. The QKD protocols can be divided into three main categories, which are called discrete-variable (DV) QKD [5,6], continuous-variable (CV) QKD [7,8], and differential-phase-shift (DPR) QKD [9,10,11], respectively. Both DV and CV systems can be integrated on chip [12,13,14,15,16] and operate at room temperature, but CV systems have significant advantages to achieve higher rate in a short distance link [17]. Thus, the CV-QKD protocols have attracted much attention in the past few years [18,19,20,21,22,23,24]. To meet a variety of application needs, much theoretical and experimental research of CV-QKD was done [25,26,27,28,29,30,31,32,33,34,35,36,37,38]. In the research of fully trusted-device protocols, it is always assumed that the devices at two legitimate partners are honest, and Eve can only control the quantum channels rather than the devices at the two parties.

However, the mismatch between practical devices and their idealized models may open security loopholes, resulting in harmful damages to the security of a protocol and the practical systems [39]. To eliminate all the loopholes of devices, fully device-independent (DI) protocols are proposed [40], which allow Eve to control all experimental devices without any assumptions. Nevertheless, DI protocols need a loophole-free Bell test [41] which is an experimental challenge. To compromise between practical protocols and full DI protocols, semi-device-independent (semi-DI) protocols are proposed, e.g., measurement-device-independent (MDI) [42,43,44], source-device-independent [45,46], and one-sided device-independent (1sDI) [47,48] QKD protocols, to consider both the security of some devices and the performance of a protocol. In semi-DI protocols, some devices can be assumed to be fully controlled by the adversary while the others should be well characterized. The investigations on the security analysis of semi-DI protocols develop very fast in recent years, such as CV-MDI [49,50,51], source-device-independent [46] and CV-1sDI protocols [47,52,53], which extend the application of such protocols.

Compared with one-way CV-QKD protocols, the secret key of CV-MDI QKD protocols is established by the measurement results of an untrusted third party, which reduces the performance of the protocols [42]. A lot of efforts were aimed at improving the performance of the protocols, such as using squeezed states [43], and virtual photon subtraction [54,55]. Recently, the unidimensional CV-QKD protocols were proposed in one-way CV-QKD protocols [56,57]. Compared with the Gaussian-modulated protocols, the features of the unidimensional CV-QKD protocols include easy modulation, low cost, and only a small number of random numbers required [56,58]. Moreover, the performance of the unidimensional coherent-state CV-QKD protocol is comparable to the Gaussian-modulated coherent-state protocol under the condition of low excess noise [56,58,59]. Even if the detectors are not ideal, the performance of the protocols can be improved by adding an optical amplifier [60]. Therefore, the unidimensional CV-QKD protocol using coherent states has a certain potential to be applied to various scenarios.

In this paper, we introduce the unidimensional CV-QKD protocol with untrusted detection under realistic conditions in order to eliminate the loopholes described above. We first present the equivalent entanglement-based (EB) scheme and the prepare-and-measure (PM) scheme of the unidimenaional CV-QKD protocol with untrusted detection under realistic conditions, including three different schemes based on using unidimensional modulated coherent states at each side (Alice’s side or Bob’s side), and both sides (both Alice’s and Bob’s side). The expressions of the secret key rate of the protocols are derived and the optimal gain parameters of the displacement are calculated, respectively. It is found that the optimal performance, in terms of both key rates and maximal transmission distance, of the protocol is achieved using unidimensional modulated coherent states at both Alice’s and Bob’s side. In addition, we also consider the asymmetric case that the distance between Bob and Charlie decreases to make the transmission distance further. Thus we investigate the relationship between the distance from Alice or Bob to Charlie by numeral simulation. Furthermore, an extreme situation is taken into account that Charlie is put on Bob’s side, and the simulation result indicates that the total transmission distance increases when the distance from Bob to Charlie decreases.

The paper is organized as follows. In Section 2, we give detailed descriptions of the PM and EB schemes of the proposed protocol in three situations. Then we derive the expressions of secret key rate in detail and show the numerical simulation results of the secret key rate. Our conclusions are drawn in Section 3.

Note added. Recently, an independent work [61] was posted on arXiv. This work studied the performance of the measurement-device-independent CV-QKD protocol using unidimensional modulated coherent states in both Alice’s and Bob’s sides.

## 2. Results

### 2.1. Unidimensional CV-QKD Protocol with Untrusted Detection

Firstly, we propose the PM scheme for unidimensional CV-QKD protocol with untrusted detection, as illustrated in Figure 1. In particular, the modulator in the model can be Gaussian modulator as well as unidimensional modulator. Thus, there are four probable situations in our discussion, among which the situation that Gaussian modulator in both sides was described in detail in references [50,51]. Therefore, the other three probable schemes are taken into consideration in the proposed protocol with unidimensional modulator, which are the unidimensional modulation only in Alice’s side, the unidimensional modulation only in Bob’s side and the unidimensional modulation both in Alice’s and Bob’s side, respectively. The PM schemes of the three cases are described separately as follows:
**Case 1: unidimensional modulation only in Alice’s side***Step 1.* Alice produces coherent states and randomly selects the x- or p-quadrature along which the prepared states are displaced according to a random Gaussian variable with displacement variance $V_{A}^{M}=V_{A}^{2}-1$. At the same time, Bob randomly prepares coherent states $|x_{B}+ip_{B}\rangle$, where $x_{B}$ and $p_{B}$ are Gaussian distributed with modulation variance $V_{B}^{M}=V_{B}-1$. Subsequently, the states are sent to the untrusted party Charlie through two different channels whose length are $L_{AC}$ and $L_{BC}$, respectively.*Step 2.* After receiving the mode $A^{\prime }$ from Alice and the mode $B^{\prime }$ from Bob, Charlie combines them with a 50:50 beamsplitter. The output are mode *C* and *D*. Subsequently, Charlie performs measurement on the x-quadrature of the mode *C* and the p-quadrature of the mode *D* with two homodyne detectors, and then announces the results $X_{C}$ and $P_{D}$ publicly through the classical channels.*Step 3.* According to the information Charlie announces, Bob modifies his data as $x_{B}^{\prime }=x_{B}+kX_{C}$, $p_{B}^{\prime }=p_{B}+kP_{D}$, where *k* is the amplification coefficient. Here Alice keeps her data unchanged.*Step 4.* Alice and Bob perform post-processing, including information reconciliation, privacy amplification, and so on.
**Case 2: unidimensional modulation only in Bob’s side***Step 1.* Alice randomly prepares coherent states $|x_{A}+ip_{A}\rangle$, where $x_{A}$ and $p_{A}$ are Gaussian distributed with modulation variance $V_{A}^{M}=V_{A}-1$. Meanwhile, Bob produces coherent states and randomly selects the x- or p-quadrature along which the prepared states are displaced according to a random Gaussian variable with displacement variance $V_{B}^{M}=V_{B}^{2}-1$. Subsequently, the states are sent to the untrusted party Charlie through two different channels whose length are $L_{AC}$ and $L_{BC}$, respectively.The next steps are the same as those in Case 1.
**Case 3: unidimensional modulation in both sides***Step 1.* Both Alice and Bob produce coherent states and simultaneously select the x- or p-quadrature along which the prepared states are displaced according to two random Gaussian variables with displacement variance $V_{A}^{M}=V_{A}^{2}-1$ and $V_{B}^{M}=V_{B}^{2}-1$, respectively. Subsequently, the states are sent to the untrusted party Charlie through two different channels whose length are $L_{AC}$ and $L_{BC}$, respectively.The next steps are the same as those in Case 1.

Furthermore, the equivalent EB schemes are described as followed, among which the Case 3 is revealed in Figure 2a:
**Case 1: unidimensional modulation only in Alice’s side***Step 1.* Alice generates Einstein-Podolsky-Rosen (EPR) states with variance $V_{A}$. Then she keeps mode $A_{1}$ and squeezes the other mode $A_{2}$ on a squeezer. The output is mode $A_{3}$, which is sent to the untrusted party Charlie through a channel with length $L_{AC}$. Meanwhile, Bob generates another Einstein-Podolsky-Rosen (EPR) state with variance $V_{B}$. Then he keeps mode $B_{1}$ and sends the other mode $B_{2}$ through a channel with length $L_{BC}$.*Step 2.* Modes $A^{\prime }$ and $B^{\prime }$ received by Charlie interfere at a 50:50 beamsplitter with two output modes *C* and *D*. Subsequently, Charlie performs measurement on the x-quadrature of the mode *C* and the p-quadrature of the mode *D* with two homodyne detectors, and then announces the results $X_{C}$ and $P_{D}$ publicly through the classical channels.*Step 3.* According to the information Charlie announces, Bob displaces mode $B_{1}$ by operation $\hat{D}\left(\beta \right)$, where $\beta =g(X_{C}+iP_{D})$, and *g* represents the gain of displacement. The relationship between *k* and *g* is well studied in reference [42]. Then Bob measures mode $B_{1}^{\prime }$ to get the final data $X_{B}$,$P_{B}$ using heterodyne detection. Alice uses mode $A_{1}$ to get the final data $X_{A}$($P_{A}$) using homodyne detection.*Step 4.* Alice and Bob perform post-processing, including information reconciliation, privacy amplification, and so on.
**Case 2: unidimensional modulation only in Bob’s side***Step 1.* Alice generates Einstein-Podolsky-Rosen (EPR) states with variance $V_{A}$. Then she keeps mode $A_{1}$ and sends the other mode $A_{2}$ through a channel with length $L_{AC}$. Meanwhile, Bob generates another Einstein-Podolsky-Rosen (EPR) state with variance $V_{B}$. Then he keeps mode $B_{1}$ and squeezes the other mode $B_{2}$ on a squeezer. The output is mode $B_{3}$, which is sent to the untrusted party Charlie through a channel with length $L_{BC}$.*Step 2* and *Step 4* are the same as those in Case 1.*Step 3.* According to the information Charlie announces, Bob displaces mode $B_{1}$ by operation $\hat{D}\left(\beta \right)$, where $\beta =g(X_{C}+iP_{D})$. Then Bob measures mode $B_{1}^{\prime }$ to get the final data $X_{B}$($P_{B}$) using homodyne detection. Alice uses mode $A_{1}$ to get the final data $X_{A}$, $P_{A}$ using heterodyne detection.
**Case 3: unidimensional modulation in both sides***Step 1.* Both Alice and Bob generate Einstein-Podolsky-Rosen (EPR) states with variance $V_{A}$ and $V_{B}$ respectively. Alice and Bob keep mode $A_{1}$ and mode $B_{1}$ of their own EPR state separately. The other two modes, $A_{2}$ and $B_{2}$, are squeezed on two squeezers, and the output are modes $A_{3}$ and $B_{3}$. Then the modes $A_{3}$ and $B_{3}$ are sent to the untrusted party Charlie through two different channels with length $L_{AC}$ and $L_{BC}$.*Step 2* and *Step 4* are the same as those in Case 1.*Step 3.* According to the information Charlie announces, Bob displaces mode $B_{1}$ by operation $\hat{D}\left(\beta \right)$, where $\beta =g(X_{C}+iP_{D})$. Then Alice measures mode $A_{1}$, Bob measures mode $B_{1}^{\prime }$ to get the final data $X_{A}$($P_{A}$), $X_{B}$($P_{B}$) using homodyne detection, respectively.

### 2.2. Security Analysis

In this section, the three schemes involved above, which are designed to reduce the cost and simplify the implementation of CV-QKD with untrusted detectors, are discussed separately. In each case, we derive the secure bound of the protocol using the EB scheme owing to ease of calculation in detail. In particular, under the assumptions that Eve controls the channels, Charlie and Bob’s EPR state, and the displacement in Figure 2a, their equivalent EB models of one-way CV-QKD model are illustrated in Figure 2b–d.

#### 2.2.1. Using Unidimensional Modulated Coherent States Only in Alice’s Side

The EB description of this case is similar to that shown in Figure 1, and the only difference is that there is no squeezer in Bob’s side. Thus, the EB scheme discussed here is equivalent to the one-way CV-QKD with unidimentional modulated coherent states and heterodyne detection shown in Figure 2b. The secret key rate *K* against collective attacks for reverse reconciliation is given by [62]
(1)$$\begin{array}{c} K^{\left(b\right)}=\beta I^{\left(b\right)}\left(A:B\right)-\chi^{\left(b\right)}\left(B:E\right), \end{array}$$ where β is the reconciliation efficiency, $I\left(A:B\right)=\frac{1}{2}log_{2}\left(\frac{V_{B}^{\left(b\right)}+1}{V_{B|A}^{\left(b\right)}+1}\right)$ is the classical mutual information between Alice and Bob, χ(B:E) is the Holevo quantity [63]:(2)$$\begin{array}{c} \chi^{\left(b\right)}\left(B:E\right)=S\left(\rho_{E}^{\left(b\right)}\right)-\sum_{x_{B}}p^{\left(b\right)}\left(x_{B}\right)S\left(\rho_{E|x_{B}}^{\left(b\right)}\right), \end{array}$$ where S(ρ) is the von Neumann entropy of the state ρ, $x_{B}$ is Bob’s measurement result obtained with the probability $p^{\left(b\right)}\left(x_{B}\right)$, $\rho_{E|x_{B}}^{\left(b\right)}$ is the corresponding state of Eve’s ancillary, and $\rho_{E}^{\left(b\right)}=\sum_{x_{B}}p^{\left(b\right)}\left(x_{B}\right)\rho_{E|x_{B}}^{\left(b\right)}$ are Eve’s partial states.

Since Eve is able to purify the whole system $\rho_{A_{1}B_{1}^{\prime }}^{\left(b\right)}$ to maximize the information she can get, we have $S\left(\rho_{E}^{\left(b\right)}\right)=S\left(\rho_{A_{1}B_{1}^{\prime }}^{\left(b\right)}\right)$. Furthermore, after Bob’s projective measurement resulting in $x_{B}$, the system $\rho_{A_{1}E}^{\left(b\right)}$ is pure, so that $S\left(\rho_{E|x_{B}}^{\left(b\right)}\right)=S\left(\rho_{A_{1}|x_{B}}^{\left(b\right)}\right)$. According to the Gaussian optimality theorem, we assume the final state $\rho_{A_{1}B_{1}^{\prime }}^{\left(b\right)}$ shared by Alice and Bob is Gaussian so that the information available to the eavesdropper is maximum [64,65]. Thus, the entropy $S(\rho_{A_{1}B_{1}^{\prime }}^{\left(b\right)})$ and $\sum_{x_{B}}p^{\left(b\right)}\left(x_{B}\right)S\left(\rho_{A_{1}|x_{B}}^{\left(b\right)}\right)$ can be calculated directly from the covariance matrices $\gamma_{A_{1}B_{1}^{\prime }}^{\left(b\right)}$ and $\gamma_{A_{1}|x_{B}}^{\left(b\right)}$. In addition, now the expression for $\chi_{BE}^{\left(b\right)}$ can be simplified as followed:(3)$$\begin{array}{c} \chi \left(B:E\right)=\sum_{i=1}^{2}G\left(\frac{\lambda_{i}-1}{2}\right)-G\left(\frac{\lambda_{3}-1}{2}\right), \end{array}$$ where $G\left(x\right)=\left(x+1\right)\log_{2}\left(x+1\right)-x\log_{2}x$, $\lambda_{1,2}$ are the symplectic eigenvalues of the covariance matrix $\gamma_{A_{1}B_{1}^{\prime }}^{\left(b\right)}$ and $\lambda_{3}$ is the symplectic eigenvalue of the covariance matrix $\gamma_{A_{1}|x_{B}^{}}^{\left(b\right)}$, which can be obtained in terms of the corresponding EB scheme. As is described in the corresponding EB scheme, mode $A_{3}$ in Alice’s side and $B_{2}$ in Bob’s side turn into mode $A^{\prime }$ and $B^{\prime }$ after the channel, which satisfy the following relationships:(4)$$\begin{array}{cc} {\hat{A}}_{x,p}^{\prime (b)} & =\sqrt{\eta_{A}}{\hat{A}}_{3x,p}+\sqrt{1-\eta_{A}}{\hat{E}}_{2x,p}, \end{array}$$(5)$$\begin{array}{cc} {\hat{B}}_{x,p}^{\prime (b)} & =\sqrt{\eta_{B}}{\hat{B}}_{2x,p}+\sqrt{1-\eta_{B}}{\hat{E}}_{5x,p}, \end{array}$$ where $\eta_{A}=10^{-\alpha L_{AC}/10}$, $\eta_{B}=10^{-\alpha L_{BC}/10}$ is the channel parameter transmittance on Alice’s and Bob’s side, with the loss of channel α=0.2 dB/km, the transmission distance between Alice and Charlie $L_{AC}$, and the transmission distance between Bob and Charlie $L_{BC}$.

Then passing through a beamsplitter, mode $A^{\prime }$ becomes mode *C* and mode $B^{\prime }$ becomes mode *D*, and
(6)$$\begin{array}{c} {\hat{C}}_{x,p}^{\left(b\right)}=\frac{1}{\sqrt{2}}\left({\hat{A}}_{x,p}^{\prime (b)}-{\hat{B}}_{x,p}^{\prime (b)}\right),{\hat{D}}_{x,p}^{\left(b\right)}=\frac{1}{\sqrt{2}}\left({\hat{A}}_{x,p}^{\prime (b)}+{\hat{B}}_{x,p}^{\prime (b)}\right). \end{array}$$

After measurement and displacement operation, mode $B_{1}$ becomes mode $B_{1}^{\prime }$, which is entangled with $A_{1}$. In addition, the relationship between mode $B_{1}$ and mode $B_{1}^{\prime }$ can be written as
(7)$$\begin{array}{c} {\hat{B}}_{1x}^{\prime (b)}={\hat{B}}_{1x}+g{\hat{C}}_{x}^{\left(b\right)},{\hat{B}}_{1p}^{\prime (b)}={\hat{B}}_{1p}+g{\hat{D}}_{p}^{\left(b\right)}, \end{array}$$ where *g* represents the gain of the displacement. Thus, the covariances of mode $A_{1}$ and mode $B_{1}^{\prime }$ in x-quadrature and p-quadrature can be calculated as
(8)$$\begin{array}{cc} \langle {\hat{A}}_{1x},{\hat{B}}_{1x}^{\prime (b)}\rangle & =\sqrt{T^{\left(b\right)}V_{A}\left(V_{A}^{2}-1\right)}, \end{array}$$
(9)$$\begin{array}{cc} \langle {\hat{A}}_{1p},{\hat{B}}_{1p}^{\prime (b)}\rangle & =-\sqrt{T^{\left(b\right)}\left(V_{A}^{2}-1\right)/V_{A}}, \end{array}$$ where $T^{\left(b\right)}=\frac{g^{2(b)}}{2}\eta_{A}$. Furthermore, the variances of mode $B_{1}^{\prime (b)}$ are calculated by
(10)$$\begin{array}{c} \begin{array}{c} V_{B_{1x}^{\prime (b)}}=V_{B}+\frac{g^{2(b)}}{2}\eta_{A}\left(V_{A}^{2}+\chi_{A}\right)+\frac{g^{2(b)}}{2}\eta_{B}\left(V_{B}+\chi_{B}\right)-g\sqrt{2\eta_{B}}\sqrt{\left(V_{B}^{2}-1\right)},\\ V_{B_{1p}^{\prime (b)}}=V_{B}+\frac{g^{2(b)}}{2}\eta_{A}\left(1+\chi_{A}\right)+\frac{g^{2(b)}}{2}\eta_{B}\left(V_{B}+\chi_{B}\right)-g\sqrt{2\eta_{B}}\sqrt{\left(V_{B}^{2}-1\right)}. \end{array} \end{array}$$

Then the covariance matrix $\gamma_{A_{1}B_{1}^{\prime }}^{\left(b\right)}$ can be written naturally as
(11)$$\begin{array}{c} \begin{array}{c} \gamma_{{A_{1}B_{1}}^{\prime }}^{\left(b\right)}=\left[\begin{array}{cccc} V_{A} & 0 & \sqrt{T^{\left(b\right)}V_{A}\left(V_{A}^{2}-1\right)} & 0\\ 0 & V_{A} & 0 & C_{p}^{\left(b\right)}\\ \sqrt{T^{\left(b\right)}V_{A}\left(V_{A}^{2}-1\right)} & 0 & T^{\left(b\right)}\left(V_{A}^{2}-1+\varepsilon^{\prime (b)}\right)+1 & 0\\ 0 & C_{p}^{\left(b\right)} & 0 & 1+T^{\left(b\right)}\varepsilon^{\prime (b)} \end{array}\right]=\left[\begin{array}{cc} \gamma_{A_{1}} & \sigma_{A_{1}B{}_{1}^{\prime }}^{T(b)}\\ \sigma_{A_{1}B{}_{1}^{\prime }}^{\left(b\right)} & \gamma_{B_{1}^{\prime }}^{\left(b\right)} \end{array}\right], \end{array} \end{array}$$ where
(12)$$\varepsilon^{\prime (b)}=\varepsilon_{A}+\frac{1}{\eta_{A}}\left[\eta_{B}\left(V_{B}+\varepsilon_{B}-1\right)+2\right]+\frac{V_{B}-1-g\sqrt{2\eta_{B}}\sqrt{V_{B}^{2}-1}}{\frac{g^{2(b)}}{2}\eta_{A}}.$$

The value of $\varepsilon^{\prime (b)}$ reaches the minimum when $g^{\left(b\right)}=\sqrt{\frac{2}{\eta_{B}}}\sqrt{\frac{V_{B}-1}{V_{B}+1}}$, and the minimum $\varepsilon^{\prime (b)}=\varepsilon_{A}+\frac{1}{\eta_{A}}\left[\eta_{B}\left(\varepsilon_{B}-2\right)+2\right]$. Furthermore, since the p-quadrature is not modulated, the correlation $C_{p}^{\left(b\right)}$ is unknown. Yet the matrix is restricted by the constraint following from Heisenberg uncertainly principle:(13)$$\gamma_{A_{1}B_{1}^{\prime }}^{\left(b\right)}+i\Omega \geq 0,$$ where $\Omega =\overset{N}{\underset{k=1}{⨁}}\omega \;and\;\omega =\left[\begin{array}{cc} 0 & 1\\ -1 & 0 \end{array}\right].$ Thus the possible values of $C_{p}^{\left(b\right)}$ is limited, and its value corresponding to the minimum secret key distribution should be concerned so that we can get the lower secure bound.

Next, the symplectic eigenvalues $\lambda_{3}$ is given by the matrix $\gamma_{A_{1}|x_{B}}^{\left(b\right)}$, which can be calculated by: (14)$$\begin{array}{c} \gamma_{A_{1}|x_{B}}^{\left(b\right)}=\gamma_{A_{1}}-\sigma_{A_{1}B_{1}^{\prime }}^{T(b)}{\left(\gamma_{B_{1}^{\prime }}^{\left(b\right)}+I\right)}^{-1}\sigma_{A_{1}B_{1}^{\prime }}^{\left(b\right)}, \end{array}$$ where *I* is an identity matrix.

#### 2.2.2. Using Unidimensional Modulated Coherent States ONLY in Bob’s Side

Similarly, the EB description in this case is similar to that shown in Figure 2a, and the only difference is that there is no squeezer in Alice’s side. In addition, the EB scheme discussed here is equivalent to the one-way CV-QKD model with homodyne detection, which is illustrated in Figure 2c. The secret key rate *K* against collective attacks for reverse reconciliation is also given by (Equation 1), where the first part in right side is now $I^{\left(c\right)}\left(A:B\right)=\frac{1}{2}log_{2}\left(\frac{V_{A}+1}{V_{A|B}^{\left(c\right)}+1}\right)$. The second part $\chi^{\left(c\right)}\left(B:E\right)$ is given by (Equation 3), where $\lambda_{1,2}^{\left(c\right)}$ are the symplectic eigenvalues of the covariance matrix $\gamma_{A_{1}B_{1}^{\prime }}^{\left(c\right)}$ and $\lambda_{3}^{\left(c\right)}$ is the symplectic eigenvalue of the covariance matrix $\gamma_{A_{1}|x_{B}}^{\left(c\right)}$. The calculations to obtain $\gamma_{A_{1}B_{1}^{\prime }}^{\left(c\right)}$ and $\gamma_{A_{1}|x_{B}}^{\left(c\right)}$ resemble those in Section 2.2.1. Finally, the matrix $\gamma_{A_{1}B_{1}^{\prime }}^{\left(c\right)}$ has the following form:(15)$$\begin{array}{c} \begin{array}{cc} \gamma_{{A_{1}B_{1}}^{\prime }}^{\left(c\right)} & =\;\left[\;\begin{array}{cccc} V_{A} & 0 & \sqrt{T_{x}^{\left(c\right)}\;\left(\;V_{A}^{2}\;-\;1\;\right)\;} & 0\\ 0 & V_{A} & 0 & C_{p}^{\left(c\right)}\\ \sqrt{T_{x}^{\left(c\right)}\;\left(\;V_{A}^{2}\;-\;1\;\right)\;} & 0 & T_{x}^{\left(c\right)}\;\left(\;V_{A}\;+\;\chi_{line_{x}}^{\left(c\right)}\;\right)\; & 0\\ 0 & C_{p}^{\left(c\right)} & 0 & T_{p}^{\left(c\right)}\;\left(\;V_{A}\;+\;\chi_{line_{p}}^{\left(c\right)}\;\right) \end{array}\;\right]\;=\left[\begin{array}{cc} \gamma_{A_{1}}^{\left(c\right)} & \sigma_{A_{1}B{}_{1}^{\prime }}^{T(c)}\\ \sigma_{A_{1}B{}_{1}^{\prime }}^{\left(c\right)} & \gamma_{B_{1}^{\prime }}^{\left(c\right)} \end{array}\right], \end{array} \end{array}$$ where $T_{x}^{\left(c\right)}=g_{x}^{2(c)}\eta_{A}/2$ and $T_{p}^{\left(c\right)}=g_{p}^{2(c)}\eta_{A}/2$. The factors $\chi_{line_{x}}^{\left(c\right)}$ and $\chi_{line_{p}}^{\left(c\right)}$ can be calculated by:(16)$$\begin{array}{c} \chi_{line_{x}}^{\left(c\right)}=\frac{1-T_{x}^{\left(c\right)}}{T_{x}^{\left(c\right)}}+\varepsilon_{x}^{\prime (c)},\chi_{line_{p}}^{\left(c\right)}=\frac{1-T_{p}^{\left(c\right)}}{T_{p}^{\left(c\right)}}+\varepsilon_{p}^{\prime (c)}, \end{array}$$ with
(17)$$\begin{array}{c} \begin{array}{c} \varepsilon_{x}^{\prime (c)}\;=\;\varepsilon_{A}\;+\;\frac{1}{\eta_{A}}\left[\eta_{B}\;\left(\;V_{B}^{2}\;+\;\varepsilon_{B}\;-\;1\;\right)\;\;+\;2\right]\;+\;\frac{V_{B}\;-\;1\;-\;g_{x}^{\left(c\right)}\sqrt{2\eta_{B}}\sqrt{V_{B}\left(V_{B}^{2}\;-\;1\right)}}{\frac{g_{x}^{2(c)}}{2}\eta_{A}},\\ \varepsilon_{p}^{\prime (c)}\;=\;\varepsilon_{A}\;+\;\frac{1}{\eta_{A}}\left[\eta_{B}\;\left(\;V_{B}^{2}\;+\;\varepsilon_{B}\;-\;1\;\right)\;\;+\;2\right]\;+\;\frac{V_{B}\;-\;1\;-\;g_{p}^{\left(c\right)}\sqrt{2\eta_{B}}\sqrt{\left(V_{B}^{2}\;-\;1\right)/V_{B}}}{\frac{g_{p}^{2(c)}}{2}\eta_{A}}. \end{array} \end{array}$$

The value of $\varepsilon_{x}^{\prime (c)}$ reaches the minimum when $g_{x}^{\left(c\right)}=\sqrt{\frac{2}{\eta_{B}}}\sqrt{\frac{V_{B}-1}{V_{B}\left(V_{B}+1\right)}}$, and the minimum $\varepsilon_{x}^{\prime (c)}=\varepsilon_{A}+\frac{1}{\eta_{A}}\left[\eta_{B}\left(\varepsilon_{B}-V_{B}-1\right)+2\right]$. The value of $\varepsilon_{p}^{\prime (b)}$ reaches the minimum when $g_{p}^{\left(c\right)}=\sqrt{\frac{2}{\eta_{B}}}\sqrt{\frac{V_{B}\left(V_{B}-1\right)}{V_{B}+1}}$, and the minimum $\varepsilon_{p}^{\prime (c)}=\varepsilon_{A}+\frac{1}{\eta_{A}}\left[\eta_{B}\left(\varepsilon_{B}-\frac{1}{V_{B}}-1\right)+2\right]$.

Also, the matrix $\gamma_{{A_{1}B_{1}}^{\prime }}^{\left(c\right)}$ is restricted by constraint following from Heisenberg uncertainly principle:(18)$$\gamma_{A_{1}B_{1}^{\prime }}^{\left(c\right)}+i\Omega \geq 0.$$

Furthermore, the symplectic eigenvalues $\lambda_{3}^{\left(c\right)}$ is given by the matrix $\gamma_{A_{1}|x_{B}}^{\left(c\right)}$, which can be calculated by: (19)$$\begin{array}{c} \gamma_{A_{1}|x_{B}}^{\left(c\right)}=\gamma_{A_{1}}^{\left(c\right)}-\sigma_{A_{1}B_{1}^{\prime }}^{T(c)}{\left(X\gamma_{B_{1}^{\prime }}^{\left(c\right)}X\right)}^{-1}\sigma_{A_{1}B_{1}^{\prime }}^{\left(c\right)}. \end{array}$$

#### 2.2.3. Using Unidimensional Modulated Coherent States Both in Alice’s and Bob’s Side

The EB description in this case is illustrated in Figure 2a, which is equivalent to the one-way CV-QKD model with homodyne detection shown in Figure 2d. Then the secret key rate *K* against collective attacks for reverse reconciliation is obtained by (Equation 1), with the $I^{\left(d\right)}\left(A:B\right)=\frac{1}{2}log_{2}\left(\frac{V_{A}}{V_{A|B}^{\left(d\right)}}\right)$. Furthermore, $\chi^{\left(d\right)}\left(B:E\right)$ is given identically by (Equation 3), and all the parameters in the expression can be obtained from the final matrix $\gamma_{A_{1}B_{1}^{\prime }}^{\left(d\right)}$, whose form is as followed:(20)$$\begin{array}{c} \begin{array}{c} \gamma_{{A_{1}B_{1}}^{\prime }}^{\left(d\right)}\;=\;\;\left[\;\begin{array}{cccc} V_{A} & 0 & \sqrt{T_{x}^{\left(d\right)}V_{A}\;\left(\;V_{A}^{2}\;-\;1\;\right)\;} & 0\\ 0 & V_{A} & 0 & C_{p}^{\left(d\right)}\\ \sqrt{T_{x}^{\left(d\right)}V_{A}\;\left(\;V_{A}^{2}\;-\;1\;\right)\;} & 0 & T_{x}^{\left(d\right)}\;\left(\;V_{A}^{2}\;-\;1\;+\;\varepsilon_{x}^{\prime (d)}\;\right)\;\;+\;1 & 0\\ 0 & C_{p}^{\left(d\right)} & 0 & 1\;+\;T_{p}^{\left(d\right)}\varepsilon_{p}^{\prime (d)} \end{array}\;\right]\;\;=\;\;\left[\;\begin{array}{cc} \gamma_{A_{1}}^{\left(d\right)} & \sigma_{A_{1}B{}_{1}^{\prime }}^{T(d)}\\ \sigma_{A_{1}B{}_{1}^{\prime }}^{\left(d\right)} & \gamma_{B_{1}^{\prime }}^{\left(d\right)} \end{array}\;\right]\;, \end{array} \end{array}$$ where $T_{x}^{\left(d\right)}=g_{x}^{2(d)}\eta_{A}/2$, $T_{p}^{\left(d\right)}=g_{p}^{2(d)}\eta_{A}/2$, and
(21)$$\begin{array}{c} \begin{array}{c} \varepsilon_{x}^{\prime (d)}=\varepsilon_{A}+\frac{1}{\eta_{A}}\left[\eta_{B}\left(V_{B}^{2}+\varepsilon_{B}-1\right)+2\right]+\frac{V_{B}-1-g_{x}^{\left(d\right)}\sqrt{2\eta_{B}}\sqrt{V_{B}\left(V_{B}^{2}-1\right)}}{\frac{g_{x}^{2(d)}}{2}\eta_{A}},\\ \varepsilon_{p}^{\prime (d)}=\varepsilon_{A}+\frac{1}{\eta_{A}}\left[\eta_{B}\left(V_{B}^{2}+\varepsilon_{B}-1\right)+2\right]+\frac{V_{B}-1-g_{p}^{\left(d\right)}\sqrt{2\eta_{B}}\sqrt{\left(V_{B}^{2}-1\right)/V_{B}}}{\frac{g_{p}^{2(d)}}{2}\eta_{A}}. \end{array} \end{array}$$

The values of $\varepsilon_{x}^{\prime (d)}$ and $\varepsilon_{p}^{\prime (d)}$ reaches the minimum similarly when
(22)$$\begin{array}{c} g_{x}^{\left(d\right)}=\sqrt{\frac{2}{\eta_{B}}}\sqrt{\frac{V_{B}-1}{V_{B}\left(V_{B}+1\right)}},g_{p}^{\left(d\right)}=\sqrt{\frac{2}{\eta_{B}}}\sqrt{\frac{V_{B}\left(V_{B}-1\right)}{V_{B}+1}}, \end{array}$$ at this time are the minimum
(23)$$\begin{array}{c} \begin{array}{cc} \varepsilon_{x}^{\prime (d)} & =\varepsilon_{A}+\frac{1}{\eta_{A}}\left[\eta_{B}\left(\varepsilon_{B}-V_{B}-1\right)+2\right],\\ \varepsilon_{p}^{\prime (d)} & =\varepsilon_{A}+\frac{1}{\eta_{A}}\left[\eta_{B}\left(\varepsilon_{B}-\frac{1}{V_{B}}-1\right)+2\right]. \end{array} \end{array}$$

Furthermore, the matrix $\gamma_{{A_{1}B_{1}}^{\prime }}^{\left(d\right)}$ is restricted by the constraint following from Heisenberg uncertainly principle:(24)$$\gamma_{A_{1}B_{1}^{\prime }}^{\left(d\right)}+i\Omega \geq 0.$$

Finally, the symplectic eigenvalues $\lambda_{3}^{\left(d\right)}$ is given by the matrix $\gamma_{A_{1}|x_{B}}^{\left(d\right)}$, which can be calculated by: (25)$$\begin{array}{c} \gamma_{A_{1}|x_{B}}^{\left(d\right)}=\gamma_{A_{1}}^{\left(d\right)}-\sigma_{A_{1}B_{1}^{\prime }}^{T(d)}{\left(X\gamma_{B_{1}^{\prime }}^{\left(d\right)}X\right)}^{-1}\sigma_{A_{1}B_{1}^{\prime }}^{\left(d\right)}. \end{array}$$

### 2.3. Numeral Simulation

In this section, the performance of the proposed three schemes of the unidimensional CV-QKD protocol with untrusted detection are illustrated and compared. In particular, we first summarize the optimal parameters of the proposed three schemes into a table illustrated in Table 1. Here, the performance of the three cases discussed above is considered to make a contrast.

The parameters that will affect the secret key rate are the reconciliation efficiency β, the variance of Alice and Bob $V_{A}$, $V_{B}$, the transmission efficiency $\eta_{A}$, $\eta_{B}$, excess noise $\varepsilon_{A}$, $\varepsilon_{B}$ of two quantum channels. It can be seen in Table 1 that the excess noises $\varepsilon_{x}^{\prime }$, $\varepsilon_{p}^{\prime }$ are related to the variance $V_{B}$. When the values of $\varepsilon_{x}^{\prime }$, $\varepsilon_{p}^{\prime }$ are less than zero, the excess noises are physically absent. Therefore, the values of variance $V_{B}$, which make the excess noises $\varepsilon_{x}^{\prime }$, $\varepsilon_{p}^{\prime }$ less than zero, are unreasonable. Conversely, when the values of $V_{B}$ make $\varepsilon_{x}^{\prime }$, $\varepsilon_{p}^{\prime }$ greater than or equal to zero at the same time, they are reasonable. Thus, we make the variance $V_{B}$ take the values 1.001, 1.1, and 2, and simulate the performances of the excess noise $\varepsilon_{x}^{\prime }$, $\varepsilon_{p}^{\prime }$. As is shown in Figure 3, when the variance $V_{B}=1.001$, the excess noises in x- and p-quadrature are both greater than zero, so we choose this value for numerical simulation. In particular, we choose a large variance of $V_{A}=10^{5}$ to see the performance of the ideal modulation, and use practical variance of $V_{A}=4$ to observe the realistic performance. Excess noise is set to $\varepsilon_{A}=\varepsilon_{B}=\varepsilon =0.001$ and transmittance are $\eta_{A}=10^{-\alpha L_{AC}/10}$, $\eta_{B}=10^{-\alpha L_{BC}/10}$ (α=0.2 dB/km) for simulation, which are standard parameters in one-way CV-QKD experiment. Furthermore, the other parameter reconciliation efficiencies in the three cases are set as β=0.98 for practical case, and β=1 for ideal case.

Firstly, we consider the performance of the symmetric case where the length of two quantum channels $L_{AC}=L_{BC}$. Then we make a numerical simulation of the secret key rates *K* in the three cases. Unfortunately, even the parameters are set to be ideal, the secret key is unable to be distilled in the case that unidimensional modulated coherent states only in Bob’s side. The phenomenon may be resulted from the structure of the scheme and the awful effect of the excess noise in the p- quadrature where the states are not modulated. Since MDI-type protocol requires displacement operation in Bob’s side, at least 1-unit extra variance will be introduced to the quadrature by Charlie’s announced data when displacing a coherent state, we find it rational that no secure key could be extracted in the case that the unidimensional modulation only in Bob’s side. Thus, the cases that unidimensional modulated coherent states only in Alice’s side as well as in both sides are taken into consideration. The simulation results are shown in Figure 4, from which we make a comparison. We find that the secret key rate of ideal condition is always larger than that of practical condition. Furthermore, it can be directly seen that the case of unidimentional modulation both in two sides corresponds to higher secret key rate and further transmission distance.

Secondly, we can consider the EB schemes of the proposed protocol as a continuous-variable quantum teleportation process, i.e., Alice and Bob prepare EPR states respectively, and then pass the mode from Alice to Bob. Therefore, any loss and noise in the channel from Bob to Charlie with the length $L_{BC}$ will reduce the quality of the EPR source, thus affecting the final performance, as is revealed in Figure 4. In other words, $L_{BC}$ has a much greater impact on the final performance than $L_{AC}$. In order to eliminate this effect as much as possible and increase the total transmission distance, we try to shorten the distance between Bob and Charlie ($L_{BC}$). The change of the total transmission distance is displayed by numerical simulation, where the distance between Bob and Charlie $L_{BC}$ is a function of the distance between Alice and Charlie $L_{AC}$. Specifically, we find the maximum $L_{BC}$, which makes the secret key rate greater than zero, corresponding to each $L_{AC}$. The results are displayed in Figure 5, from which we can find that when Charlie’s position is close to Bob, the total maximal transmission distance $L_{AB}$ ($L_{AB}=L_{AC}+L_{BC}$) will be relatively longer. Also, $L_{AB}$ improves with large variance $V_{A}=10^{5}$. Furthermore, examining different locations the unidimentional modulation in, we find that the identical result that the transmission distance corresponding to the unidimentional modulation in both sides has a better performance.

Finally, an extreme asymmetric situation is considered when $L_{BC}=0$. As revealed in Figure 6, the transmission distance between the two legitimate parties $L_{AB}$ increases significantly in a comparison with the symmetric case. In this case, the secret key rates correlated with the unidimentional modulation in both sides provide a better performance and the performance of the secret key rate in the ideal condition is better than that in the practical condition. Besides, we also plot the curves of standard CV-MDI QKD in Figure 6 for a better understanding of the secret key rate performance of our proposed protocol. As is revealed in Figure 6, the performance of the proposed protocol is comparable to the standard CV-MDI QKD protocol.

## 3. Discussion and Conclusions

In this paper, a unidimensional continuous-variable quantum key distribution protocol with untrusted detection under realistic conditions is proposed. We consider three situations including using unidimensional modulated coherent states at each side or both sides and derive the expressions of the secret key rates against the collective attacks of protocols in each situation, where the third party is untrusted and may be controlled by the eavesdropper. Making use of the expression we make numeral simulations and compare the performances of the cases that the unidimensional modulation exists only in Alice’s side as well as in both sides. From the simulation results can we know that the protocol provides a better performance when the unidimensional modulation is used in both sides of the two legitimate partners, and decreasing the distance between Bob and Charlie helps make the total transmission distance further. Indeed, with the appropriate parameters and schemes selected, we could extract the secret key based on the proposed protocol except in the case of unidimensional coherent states only in Bob’s side. We provide a possible explanation of the phenomenon, and the reason for more accuracy is still an open question. We would like to model this situation better in the future. Undoubtedly, the proposed protocol provides a simple method to simplify the implementation of the CV-QKD systems, and the security analysis is based on the uncertainty relation. In addition, the scheme has the ability to immune to the collective attacks against standard detectors that are very likely to exist in practical system. 

## Figures and Tables

**Figure 1 entropy-21-01100-f001:**
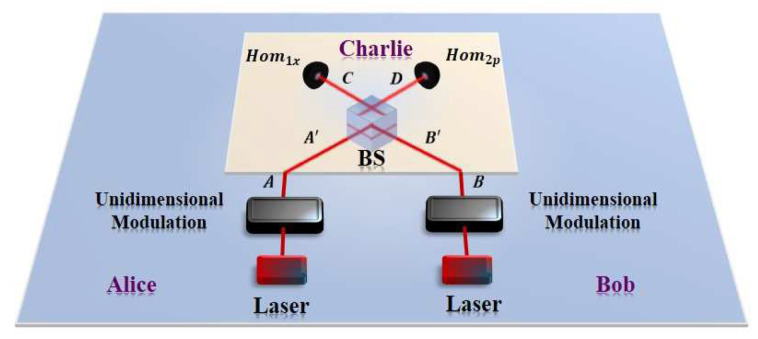
(Color online) The PM scheme of the unidimensional CV-QKD protocol with untrusted detection with the unidimensional modulator in both sides, where both Alice and Bob perform unidimensional modulation. Replacing the unidimensional modulation in Bob’s side with the standard Gaussian modulation corresponds to the case of unidimensional modulation only in Alice’s side, while replacing the unidimensional modulation in Alice’s side with the standard Gaussian modulation corresponds to the case of unidimensional modulation only in Bob’s side. In particular, the quantum channels and Charlie are fully controlled by Eve.

**Figure 2 entropy-21-01100-f002:**
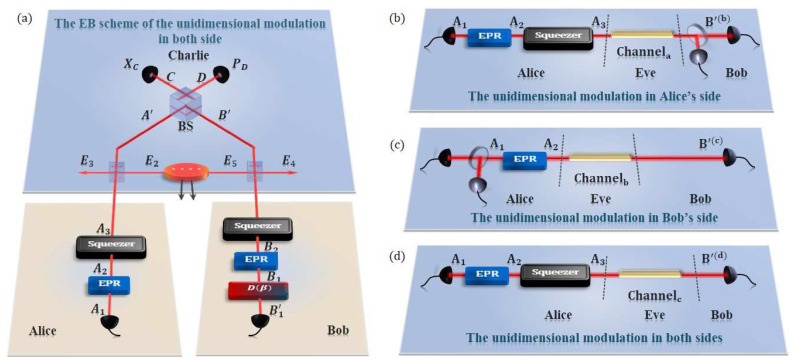
(Color online) The EB scheme and the equivalent one-way model of the unidimensional CV-QKD protocol, where the EPR states are two-mode vacuum states, with untrusted detection and coherent states (**a**) The EB scheme of the unidimentional modulation both in Alice’s and Bob’s side where the detectors are all homodyne detector. (**b**) The equivalent one-way model of the case that are the unidimentional modulation only in Alice’s side. (**c**) The equivalent one-way model of the case that are the unidimentional modulation only in Bob’s side. (**d**) The equivalent one-way model of the case that are the unidimentional modulation both in Alice’s and Bob’s side. In particular, two quantum channels and Charlie are fully controlled by Eve, but Eve has no access to the apparatuses in Alice’s and Bob’s stations.

**Figure 3 entropy-21-01100-f003:**
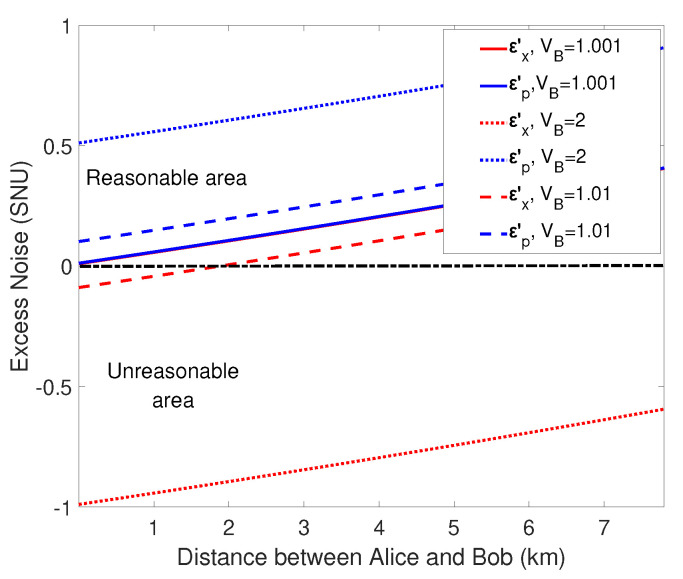
(Color online) Excess noise versus distance with different $V_{B}$ in the situation that unidimensional modulation is in both sides. The dotted lines are under the condition that $V_{B}=2$, the dashed lines represent the condition that $V_{B}=1.1$, and the solid lines represent the case that $V_{B}=1.001$. $\varepsilon_{A}$ and $\varepsilon_{B}$ is set as $\varepsilon_{A}=\varepsilon_{B}=0.001$. In particular, the region is divided into two parts by a black dotted-dashed line, where upper part is a reasonable region, indicating that the excess noise is greater than zero, and the lower half is an unreasonable area, indicating that the excess noise is less than zero.

**Figure 4 entropy-21-01100-f004:**
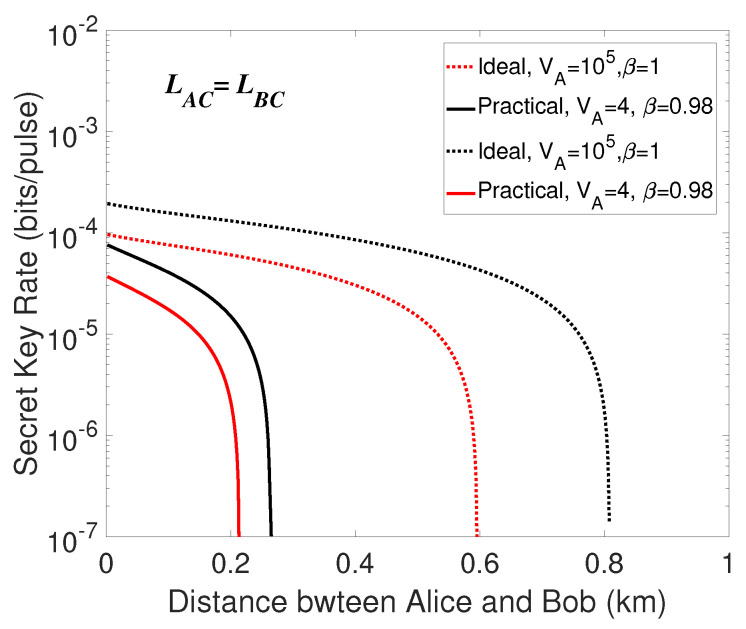
(Color online) Secret key rate in the symmetric case ($L_{AC}=L_{BC}$). the dotted lines are under the ideal condition ($V_{A}=10^{5}$, β=1 in the situations of unidimensional modulation only in Alice’s side as well as in both sides) and the solid lines represent the practical condition ($V_{A}=4$, β=0.98 in the situations of unidimensional modulation only in Alice’s side as well as in both sides). The red lines represent the case that the unidimentional modulation only exists in Alice’s side, the black lines are on behalf of the case that the unidimensional modulation exists in both sides.

**Figure 5 entropy-21-01100-f005:**
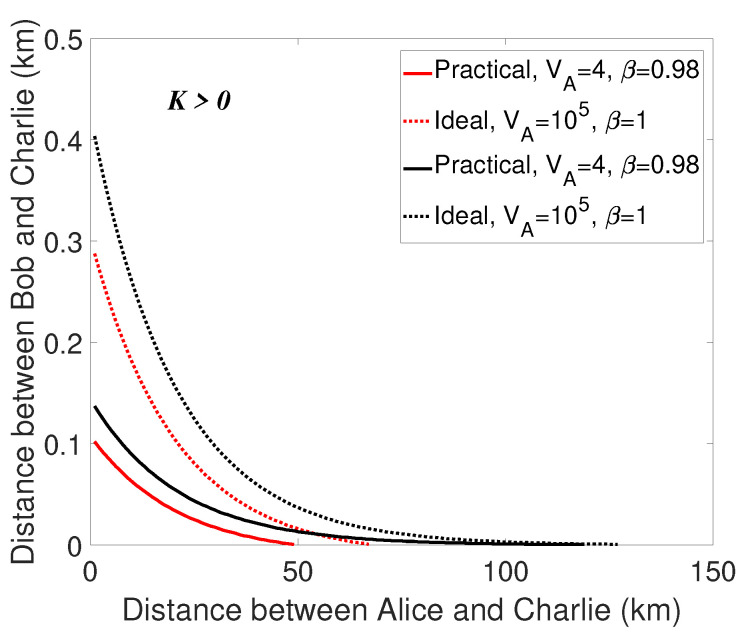
(Color online) Curves of the correlation between $L_{AC}$ and $L_{BC}$. The transmission distance from Alice and Charlie $L_{AC}$ is considered to be a function of the distance from Bob to Charlie $L_{BC}$. the dotted lines are under the ideal condition ($V_{A}=10^{5}$, β=1 in the situations of unidimensional modulation only in Alice’s side as well as in both sides) and the solid lines represent the practical condition ($V_{A}=4$, β=0.98 in the situations of unidimensional modulation only in Alice’s side as well as in both sides). The red lines represent the case that the unidimentional modulation only exists in Alice’s side, the black lines are on behalf of the case that the unidimensional modulation exists in both sides of Alice and Bob. The excess noise to be $\varepsilon_{A}=\varepsilon_{B}=0.001$.

**Figure 6 entropy-21-01100-f006:**
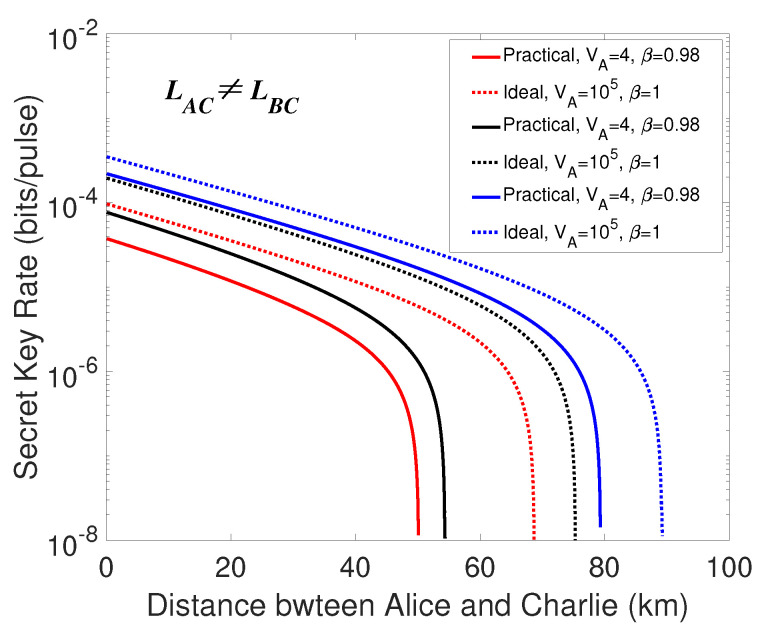
(Color online) Secret key rate versus transmission distance between Alice and Charlie, and the distance $L_{BC}$ is set to $L_{BC}=0$. Identically, the dotted lines are under the ideal condition, and the solid lines represent the practical condition. The red lines represent the case that the unidimentional modulation only exists in Alice’s side, the black lines are on behalf of the case that the unidimensional modulation exists in both sides of Alice and Bob, and the blue lines are the secret key rates of the standard CV-MDI protocols. The parameters are set the same as the symmetric situation.

**Table 1 entropy-21-01100-t001:** Optimal parameters of the unidimensional CV-QKD protocol with untrusted detection.

	Using Unidimensional Modulated Coherent States Only in Alice’s Side	Using Unidimensional Modulated Coherent States Only in Bob’s Side	Using Unidimensional Modulated Coherent States Only in Both Sides
$\varepsilon_{x}^{\prime }$	$\varepsilon_{A}+\frac{1}{\eta_{A}}\left[\eta_{B}\left(\varepsilon_{B}-2\right)+2\right]$	$\varepsilon_{A}+\frac{1}{\eta_{A}}\left[\eta_{B}\left(\varepsilon_{B}-V_{B}-1\right)+2\right]$	$\varepsilon_{A}+\frac{1}{\eta_{A}}\left[\eta_{B}\left(\varepsilon_{B}-V_{B}-1\right)+2\right]$
$\varepsilon_{p}^{\prime }$	$\varepsilon_{A}+\frac{1}{\eta_{A}}\left[\eta_{B}\left(\varepsilon_{B}-2\right)+2\right]$	$\varepsilon_{A}+\frac{1}{\eta_{A}}\left[\eta_{B}\left(\varepsilon_{B}-\frac{1}{V_{B}}-1\right)+2\right]$	$\varepsilon_{A}+\frac{1}{\eta_{A}}\left[\eta_{B}\left(\varepsilon_{B}-\frac{1}{V_{B}}-1\right)+2\right]$
$g_{x}$	$\sqrt{\frac{2}{\eta_{B}}}\sqrt{\frac{V_{B}-1}{V_{B}+1}}$	$\sqrt{\frac{2}{\eta_{B}}}\sqrt{\frac{V_{B}-1}{V_{B}\left(V_{B}+1\right)}}$	$\sqrt{\frac{2}{\eta_{B}}}\sqrt{\frac{V_{B}-1}{V_{B}\left(V_{B}+1\right)}}$
$g_{p}$	$\sqrt{\frac{2}{\eta_{B}}}\sqrt{\frac{V_{B}-1}{V_{B}+1}}$	$\sqrt{\frac{2}{\eta_{B}}}\sqrt{\frac{V_{B}\left(V_{B}-1\right)}{V_{B}+1}}$	$\sqrt{\frac{2}{\eta_{B}}}\sqrt{\frac{V_{B}\left(V_{B}-1\right)}{V_{B}+1}}$

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
