# Peer review of "Unidimensional Continuous-Variable Quantum Key Distribution with Untrusted Detection under Realistic Conditions"

_entropy, 2019, doi:10.3390/e21111100_

Round 1
Reviewer 1 Report
The paper “Unidimensional continuous-variable quantum key distribution with untrusted detection under realistic conditions” presents a theoretical investigation of unidimensional continuous-variable quantum key distribution system. The work appears novel and interesting. The theory is documented with rigour and the results seem to be significant progress. As such, I consider it suitable for publication in Entropy after major revisions. However, there are some points that need to be clarified.
Some minor points on the paper which can improve the final readability:
The first introduction paragraph must be changed:
“The QKD protocol can be classified into three main classed DV-QKD and CV-QKD and DPR-QKD systems, respectively.” Please include these citations. It’s important to properly define the QKD protocols.
- Scarani, V. et al., The security of practical quantum key distribution. Rev. Mod. Phys. 81, 1301-1350 (2009);
Inoue, K., Waks, E. & Yamamoto, Y. Differential-phase-shift quantum key distribution using coherent light. Phys. Rev. A 68, 022317 (2003). Stucki, D., Brunner, N., Gisin, N., Scarani, V. & Zbinden, H. Fast and simple one-way quantum key distribution. Appl. Phys. Lett. 87, 194108 (2005). Bacco D. et al., Two-dimensional distributed-phase-reference protocol for quantum key distribution, Reports 6:36756 (2016)In the introduction again, the authors reported: “The continuous-variable approach of QKD 20 (CV-QKD) protocol [3,4] has attracted much attention in the past few years mainly because it offers the advantages over discrete-variable systems of higher secret key rates in metropolitan areas, the use of standard telecom components that can operate at room temperature, and the potential for being integrated and miniaturized”
I disagree with what the authors said: it’s true that CV approach promise to achieve higher rate in a short distance link, but both DV and CV can use room temperature detection and can be integrated on chip and they can also be multiplexed. Please have a look on these references and include them in the main text:
Sibson, Philip, et al. "Chip-based quantum key distribution." Nature communications8 (2017): 13984. Ding, Yunhong, et al. "High-dimensional quantum key distribution based on multicore fiber using silicon photonic integrated circuits." npj Quantum Information1 (2017): 25. Agnesi, C., et al. "Hong–Ou–Mandel interference between independent III–V on silicon waveguide integrated lasers." Optics letters2 (2019): 271-274. Bacco, Davide, et al. "Space division multiplexing chip-to-chip quantum key distribution." Scientific reports1 (2017): 12459.
The setup figures are clear and the theory reported is correct. However, the rest of the figures are not clear to me. Example in Figure 3. Does it mean that Alice and Bob must be located very close? If yes how? We are expecting to deal with a practical implementation of the protocol, so the authors should provide more info on this side.
Figure 4. Missing x axis notation, meter, kilometer? What does it mean reasonable? Reasonable in my opinion is not a scientific term, so please provide a better explanation on this point.
Figure 5. Again not very clear plot.
Figure 6. Ok, this plot is fine and correct but a comparison with a more classical-gaussian modulation CV-MDI QKD is required for a better understanding of the secret key rate performance.
Conclusion and discussion. I would expect a deeper discussion about future realization of this protocol. Where is worth to use this protocol compared to other protocols?
Reviewer 2 Report
The paper introduces a new CV-QKD protocol using and untrusted part.
Directly to the point of concern:
1) The protocol is not properly presented. This is crucial for the reader to understand are the steps of both Alice Bob and Charlie, their public, private knowledge and output.
2) Also, in the line of 1), the EB version must be clear for the reader.
3) In the security analysis, the authors must explain why the 3 schemes (and the meaning of the scheme). In my opinion a scheme is a possible scenario of attack. If this is the case then, why this 3 cases saturate all the others possible attacks.
4) No comparison of key-rate with other proposals with CV (or even others) are discussed. This is relevant to infer the usage of the protocol in the real world when compared with other proposals.
5) I did not check carefully the detail of the calculations. I am assuming that the authors made a carefully check of all the derivations.
6) In the discussion the authors talk en passant about some of the results. For example, line 155 the value V_B is said to have suitable effect (for what?) and therefore is is used for simulations? What are the results for the worst case? Furthermore, line 163, since you use ideal case, you are not able to distill the unidimensional case. So, why is this case useful in the discussion, and what be reasonable to deviate from the protocol, or the level of tolerable noise that would allow it t be distilled?
7) Finally, and also in the discussion, it is stated that the relevant results are1) the ideal case performs better than the practical one. (This is more than expected). 2) The distance between Bob and Charlie are the one that matters. I would expect the protocol to be symmetric and therefore not only the distance to Bob, but also the distance to Alice, to have influence in the performance. So, this second conclusion deserves more explanation and detail.
Reviewer 3 Report
In this article the authors evaluate the performance of a continuous variable quantum key distribution protocol. According to this protocol, unidimensional modulated or Gaussian modulated coherent states from Alice and Bob are sent to an untrusted agent, Charlie. The authors provide expressions for the secret key rates for the cases where unidimensional modulation is used in one or both sides of communication. Comparing the results obtained, they conclude that the best performance is achieved when unidimensional modulation is used in both Alice and Bob sides.
The work seems to be scientifically sound and the research topic is timely and interesting with obvious potential applications. For these reasons we believe that the submitted article warrants publication in Entropy, after the authors improve the English of the manuscript.
Reviewer 4 Report
In the manuscript under consideration authors study a what seems to be measurement-device-independent (MDI) CV-QKD protocol using unidimensional (UD) quadrature coding of coherent signal states. The study covers a missing puzzle in the literature, considers three possible scenarios, is scientifically sound and generally well explained, hence the manuscript can be accepted for publication once authors consider the comments and suggestions below.
1. The title of the manuscript and it's positioning in the introduction are somewhat confusing. The authors seem to study MDI UD CV QKD unidimensional protocol (giving untrusted interferometric detection to a third party), but instead they call it UD CV0QKD protocol with untrusted detection, which could also refer to a conventional CV QKD under "paranoid assumption" on untrusted measurement device at Bob's station. Therefore authors can consider being more specific in relating their study to existing methods.
2. MDI CV QKD was recently studied in arXiv:1905.09029. Although the paper doesn't seem to be published yet (indeed, DOI on arXiv points at another paper), authors should cite it and comment on similarities or differences in the used approach and obtained results.
3. "as one of most prominent application of quantum information science" -> "is one of most prominent applications of quantum information science"
4. When discussing "fully trusted-device protocols," authors can cite a review
5. The phrase "However, because the mismatch between practical devices and their idealized models may open security loopholes, which make the practical systems vulnerable to attacks, and it compromise the security of a protocol" seems unfinished
6. "experimental challenging" -> "experimentally challenging"
7. "there are four probable situation" -> "there are four probable situations"
8. "calculated directly by the covariance matrices" -> "calculated directly from the covariance matrices"
9. "which can be obtained making a refer to the scheme in Fig. 2" - consider rewriting
10. Equations (4,5) describing, as authors claim, modes evolution, are confusing. They rather describe evolution of quantum operators of respective modes and should be explained accordingly, similarly for (6-9)
11. "Further, the variances <...> is calculated " -> "Further, the variances <...> are calculated "
12. In the caption to Fig. 3 authors refer to the case \beta=1 as the practical condition, while \beta in practice should be also limited, same for Fig. 5.
13. It is confusing to observe negative excess noise in Fig. 4. Authors should clearly explaining what is plotted here and why noise turns to negative values. In principle, plots of the excess noise below zero do not make sense.
14. "We consider three situation" -> "We consider three situations"
15. "proposes a new and simple method to simplify the implementation" - please consider rewriting
16. "withn" -> "with"
17. Authors mention in conclusion that "the experimental implementation of the proposed protocol faces many challenges", it would be great to explain which ones, in particular.
18. Authors should cite recent reviews on QKD (npj QI 2, 16025, 2016; arXiv:1906.01645)
Round 2
Reviewer 1 Report
After the revision, I will suggest the publication of the paper.
Author Response
Thank you.
Reviewer 2 Report
I am happy with the answers of the authors. I would just ask them to add a sentence in the conclusions point out as future work the question regarding the accuracy mentioned in the answer to point 6.
